# Insights into the modulation of bacterial NADase activity by phage proteins

Hang Yin[1,2,7], Xuzichao Li[1,3,7], Xiaoshen Wang[1,3,7], Chendi Zhang[4,7], Jiaqi Gao[1,3], Guimei Yu [1], Qiuqiu He[1,3], Jie Yang [1], Xiang Liu [5], Yong Wei[6] ✉, Zhuang Li [4] ✉ & Heng Zhang [1,3] ✉

The Silent Information Regulator 2 (SIR2) protein is widely implicated in antiviral response by depleting the cellular metabolite NAD⁺. The defense-associated sirtuin 2 (DSR2) effector, a SIR2 domain-containing protein, protects bacteria from phage infection by depleting NAD⁺, while an anti-DSR2 protein (DSR anti-defense 1, DSAD1) is employed by some phages to evade this host defense. The NADase activity of DSR2 is unleashed by recognizing the phage tail tube protein (TTP). However, the activation and inhibition mechanisms of DSR2 are unclear. Here, we determine the cryo-EM structures of DSR2 in multiple states. DSR2 is arranged as a dimer of dimers, which is facilitated by the tetramerization of SIR2 domains. Moreover, the DSR2 assembly is essential for activating the NADase function. The activator TTP binding would trigger the opening of the catalytic pocket and the decoupling of the N-terminal SIR2 domain from the C-terminal domain (CTD) of DSR2. Importantly, we further show that the activation mechanism is conserved among other SIR2-dependent anti-phage systems. Interestingly, the inhibitor DSAD1 mimics TTP to trap DSR2, thus occupying the TTP-binding pocket and inhibiting the NADase function. Together, our results provide molecular insights into the regulatory mechanism of SIR2-dependent NAD⁺ depletion in antiviral immunity.

The Sirtuin (SIR2) family proteins are widely distributed from prokaryotes to eukaryotes. Since initially discovered in yeast[1], SIR2 family proteins have been demonstrated to function as NAD⁺-dependent protein deacetylases in eukaryotes, which are involved in regulating gene expression, recombination, inflammation, and aging[2–6]. The SIR2 domains in bacteria are usually fused with other functional domains, such as APAZ (analog of PAZ), SLOG (SMF/DprA and LOG), and STAND (signal transduction ATPases with numerous associated domains)[7–12]. NAD⁺ is crucial for cellular metabolism[13–15]. Therefore, depleting NAD⁺ would affect the phage replication and/or induce host cell arrest, ultimately leading to the inhibition of phage propagation. In recent years, there has been increasing evidence to suggest that the SIR2-

[1]State Key Laboratory of Experimental Hematology, Key Laboratory of Immune Microenvironment and Disease (Ministry of Education), The Province and Ministry Co-sponsored Collaborative Innovation Center for Medical Epigenetics, International Joint Laboratory of Ocular Diseases (Ministry of Education), Tianjin Key Laboratory of Ocular Trauma, School of Basic Medical Sciences, Tianjin Medical University, Tianjin, China. [2]Department of Pharmacology, School of Basic Medical Sciences, Tianjin Medical University, Tianjin, China. [3]Department of Biochemistry and Molecular Biology, School of Basic Medical Sciences, Tianjin Medical University, Tianjin, China. [4]State Key Laboratory of Biocatalysis and Enzyme Engineering, School of Life Sciences, Hubei University, Wuhan, China. [5]State Key Laboratory of Medicinal Chemical Biology, Frontiers Science Center for Cell Responses, College of Life Sciences, Nankai University, TianJin, China. [6]The Cancer Hospital of the University of Chinese Academy of Sciences (Zhejiang Cancer Hospital), Institute of Basic Medicine and Cancer (IBMC), Chinese Academy of Sciences, HangZhou, China. [7]These authors contributed equally: Hang Yin, Xuzichao Li, Xiaoshen Wang, Chendi Zhang. ✉e-mail: weiyong@ibmc.ac.cn; zhuangli@hubu.edu.cn; zhangheng134@gmail.com

containing fusion proteins provide defense against phage by the SIR2-dependent NAD⁺ depletion. For instance, a SIR2 domain-containing protein, SIR2-APAZ, forms a stable complex with a short prokaryotic Argonaute (pAgo) protein[7]. Upon binding to the phage DNA, the SIR2 domain is dissociated from this heterodimeric SIR2-APAZ/Ago complex, thereby unleashing the NADase activity of the SIR2 domain[16,17]. Similarly, the ThsA protein from the Thoeris system, which has an N-terminal SIR2 domain and a C-terminal SLOG domain, fights against phage by depleting cellular NAD⁺[10]. The catalytic pocket of the SIR2 domain in ThsA is opened after the SLOG domain senses the cyclic ADP-ribose molecules produced by ThsB[11,18].

Most recently, defense-associated sirtuin (DSR) proteins, featuring a SIR2 domain at their N-terminus, have been found to be prevalent in bacteria[8,12]. The DSR protein family can be divided into two systems (DSR1 and DSR2). Despite limited sequence homology between the C-terminal regions of DSR1 and DSR2, both of which exert measurable anti-phage activity, which is attributed to the N-terminal SIR2 domains[8]. However, some phages have evolved the anti-DSR2 protein, such as DSAD1 (DSR anti-defense 1), to inhibit the enzymatic activity of DSR2[12]. A tail tube protein (TTP) from phage SPR, which belong to the *Siphoviridae* family[19], is further identified to trigger the NADase function of DSR2, leading to abortive infection[12]. Nonetheless, the detailed molecular mechanisms of DSR2-mediated anti-phage defense are still unclear.

Here, we present multiple cryo-EM structures of the DSR2 system in different states. DSR2 oligomerizes into an elongated assembly by the simultaneous dimerization of the C-terminal domains and tetramerization of the N-terminal SIR2 domains, supporting the emerging paradigm of cooperative self-assembly formation in immune signaling pathways[20–22]. Combined with biochemical and mutagenesis studies, we uncover the mechanisms underlying TTP recognition and NADase activation. Importantly, we also confirm that the activation model is applied to other SIR2 defense systems. More interestingly, the inhibitor DSAD1 is revealed to inactivate DSR2 by mimicking TTP. Our work builds the foundation for further mechanistic characterization of SIR2-dependent NAD⁺ depletion in bacterial immunity.

## Results

### Cryo-EM structure of DSR2-DSAD1 complex
Defense-associated sirtuin 2 (DSR2) protein has been reported to defend against phage through its NADase enzymatic activity[8,12]. The DSR2 protein (WP_029317421) from *Bacillus subtilis* 29R contains an N-terminal NADase SIR2 domain (aa 1–303) and a C-terminal domain of unknown function (aa 304–1005, termed CTD), which is predicted to be predominantly composed of α-helices[23] (Fig. 1a). We expressed and purified the DSR2 protein. Native gel electrophoresis suggested that DSR2 mainly behaves as a tetramer in solution (Supplementary Fig. 1a), which is further confirmed by the analytical ultracentrifugation (AUC) experiments (Supplementary Fig. 1b). In contrast, the SIR2 domain-only protein (aa 1–303) exists as a monomer in solution (Supplementary Fig. 1c). To understand the assembly mechanism, we sought to solve the structure of DSR2 alone by cryo-EM. However, we could not obtain a high-resolution 3D reconstruction, possibly due to intrinsic flexibility (2D average will be discussed below in Supplementary Fig. 4a). Given that the phage encoding protein DSAD1 (WP_004399562) has been shown to bind and inhibit DSR2[12], we hypothesized that the DSAD1 binding might stabilize DSR2. Our AUC results further demonstrated that DSAD1 binding does not alter the assembly status of DSR2 (Supplementary Fig. 1d, e). We therefore reconstituted the DSR2-DSAD1 binary complex and determined the cryo-EM structures at resolutions of 3.4–3.6 Å (Fig. 1b and Supplementary Figs. 2, 3) (Supplementary Table 1). We modeled four DSR2 molecules and two DSAD1 molecules into the EM density. Nearly all the residues could be clearly built except for some loop regions, including aa 1–7, 567–577, and 897–908 for DSR2 and aa 1–9 for DSAD1.

### The SIR2 domain assembly mediates the DSR2 tetramer formation
The tetrameric DSR2 displays an elongated shape, with two DSR2 dimers forming a head-to-head tetramer (the four molecules are termed A, B, A′, and B′, respectively) (Fig. 1b). The two-dimensional (2D) class averages of DSR2 alone showed extended DSR2 particles similar to those of the DSR2-DSAD1 complex (Supplementary Fig. 4a), implying that the binding to DSAD1 does not substantially alter the conformation of DSR2.

Although the SIR2 domain-only protein is a monomer (Supplementary Fig. 1c), the four SIR2 domains oligomerize into a tetramer in the context of full-length DSR2, facilitating the assembly of DSR2. The SIR2 tetramer is organized as a dimer of dimers with dihedral D2 symmetry (Supplementary Fig. 4b). The N-terminal SIR2 domain contains a Rossmann-fold subdomain and a helical lid subdomain (aa 56–111), the cleft between which is the potential NAD⁺ binding site (Fig. 1c and Supplementary Fig. 4c). The lid region of SIR2 is primarily responsible for the inter-dimer interaction. At the inter-dimer interface, Tyr71 in the lid region is sandwiched between Glu254 and Glu256 from the neighboring dimer (Fig. 1d). Moreover, a hydrogen bond is established between Tyr71 and Thr257. Three residues (Arg86, Gln89, and Ile90) of the lid region stack against Tyr260 from the adjacent dimer, further stabilizing the inter-dimer association.

### The interface of DSR2 dimer
The two-fold symmetric DSR2 dimer adopts an X-shaped architecture (Fig. 1b and Supplementary Fig. 5a). Two DSR2 molecules dimerize through both the SIR2 and CTD domains: the two C-terminal helical domains intersect at the central position, with SIR2 domains capped at the ends. Within the DSR2 dimer, the face-to-face SIR2 dimer interface is formed by the α9, α11, and α12-β6 loop (Supplementary Fig. 5a, b), predominantly stabilized by hydrophobic interactions (Supplementary Fig. 5b). In comparison with the SIR2-SIR2 dimerization interface, it appears that the CTD contributes more to the dimeric assembly, with a buried surface area of ~2200 Å². The CTD, which exhibits a fishhook-like conformation, can be divided into four helical subdomains, termed H1 (aa 304–403), H2 (aa 404–550), H3 (aa 551–860), and H4 (aa 861–1005) subdomains (Fig. 1c), and the dimeric DSR2 formation involves H2-H4 subdomains (Supplementary Fig. 5a). The H4 subdomain, the throat part of the fishhook, packs face-to-face against the other H4 subdomain from the adjacent molecule via their two C-terminal α-helices. The H4-involved dimeric assembly is driven by both hydrophobic and polar contacts (Supplementary Fig. 5c). The two H3 subdomains in the dimer corresponding to the shank part of a fishhook make a crossover, and the H3-involved dimer interface consists of a six-helix bundle with three α-helices from each molecule (Supplementary Fig. 5d). Notably, the H2 subdomain engages the SIR2 domain from the adjacent molecule, contributing to the stabilization of the dimerization (to be discussed below).

### DSAD1 binds to the H3 and H4 subdomains
DSAD1 acts as a monomer in solution (Supplementary Fig. 1d). The DSAD1 in DSR2-bound complex is composed of eight β-strands and one α-helix (Supplementary Fig. 6a). The N-terminal β-hairpin, along with the following three-stranded antiparallel β-sheet (termed sheet-1), wraps around the α-helix. Another β-sheet (termed sheet-2), consisting of three β-strands, is packed adjacent to the β-hairpin. For this fold, no structural homologs were identified for DSAD1 using the DALI server[24].

DSAD1 is trapped by the tip of the CTD fishhook (Fig. 1b, c). Particularly, sheet-2 of DSAD1 packs above the H3 subdomain (Fig. 2a, b). The H4 subdomain, along with the H3 subdomain from the adjacent molecule, brackets the β-hairpin of DSAD1 (Fig. 2c). Notably, Tyr16 of DSAD1 stacks against Tyr574 and Phe576 from the adjacent H3 subdomain (Fig. 2c). The interaction between DSR2 and DSAD1 was significantly compromised when mutating Tyr574 and Phe576 to

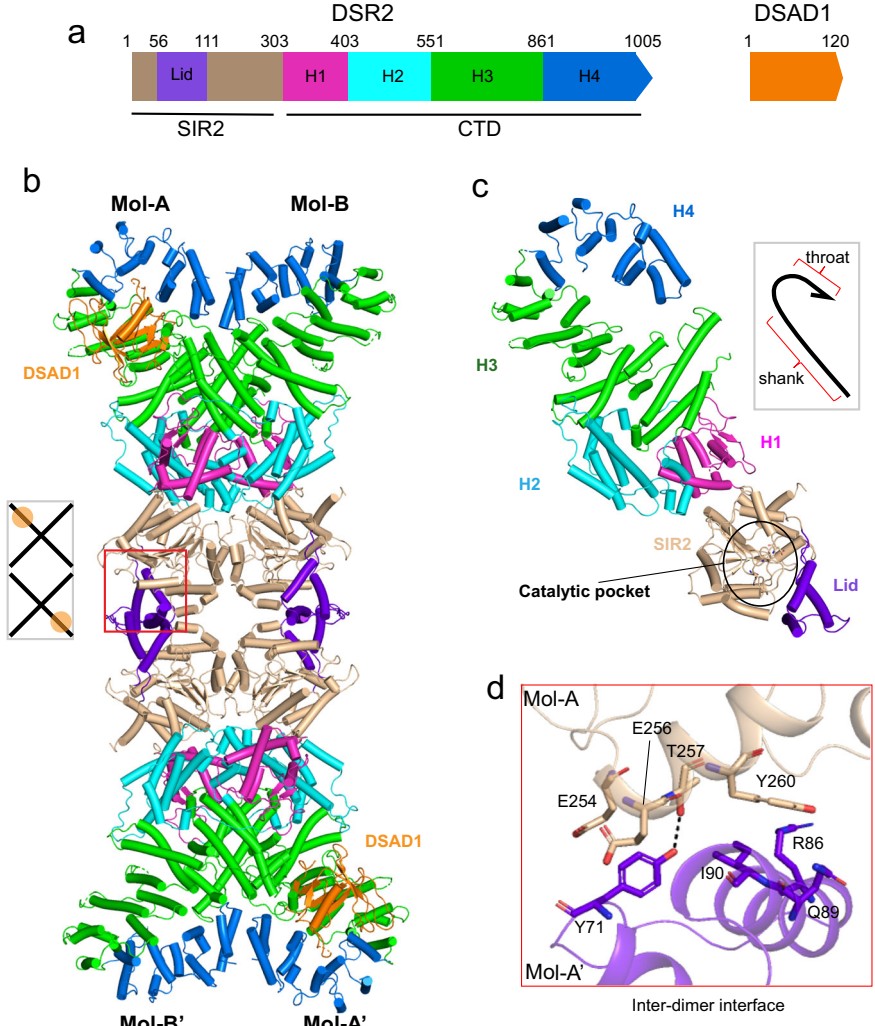

**Fig. 1 | Cryo-EM structure of DSR2 in complex with DSAD1. a** Domain organization of the DSR2 and DSAD1 proteins. **b** Cryo-EM structure of the DSR2-DSAD1 binary complex. The N-terminal SIR2 domain, Lid subdomain, and C-terminal H1, H2, H3, and H4 subdomains are colored in wheat, purple, magenta, cyan, green and blue, respectively. The DSAD1 protein is colored in orange. The lid region of the SIR2 domain responsible for the inter-dimer interaction is marked in the red box. The schematic diagram in the gray box shows the binding mode of DSAD1 to DSR2 molecules. **c** Overall structure of a single DSR2 molecule indicates a fishhook-like architecture (gray box). The catalytic pocket within DSR2 is marked in a black circle. The same color scheme as in (**a**) is used. **d** Detailed insights into the inter-dimer interface formed by the SIR2 domains. Key residues involved in the inter-dimer interaction are shown in stick representation.

glycine (Fig. 2d), indicating that the formation of DSR2 dimer appears to be important for DSAD1 binding.

While the overall structure of the DSR2 molecule in the absence of DSAD1 is similar to that of the DSAD1-bound DSR2 molecule (Supplementary Fig. 6b), subtle structural discrepancies are discerned within the H4 subdomain. The H4 subdomain moves outward by ~10 Å to accommodate DSAD1 (Supplementary Fig. 6c). The H4 subdomain seems to be critical for DSAD1 binding, as supported by the fact that the H4 subdomain and DSAD1 constitute the largest interface between DSR2 and DSAD1. Furthermore, DSAD1 was compromised in binding to DSR2 protein upon the removal of the H4 subdomain (Fig. 2d).

Although each DSR2 dimer binds to one DSAD1 molecule, two binding modes are found in the context of the DSR2 tetramer. One class of particles shows extra densities corresponding to DSAD1 on the same side (Fig. 1b and Supplementary Fig. 2a). In other words, the two DSAD1 molecules bind to DSR2-A and DSR2-A', respectively. The second binding mode involves DSR2-A and DSR2-B' molecules (Supplementary Figs. 2a and 6d). However, these DSR2-DSAD1 trimeric subcomplexes in the two states are nearly identical (Supplementary Fig. 6e).

### Activation of DSR2 by the tail tube protein

Neither full-length DSR2 nor SIR2 domain-only proteins exhibited obvious NADase activity, as determined by the ε-NAD$^+$ degradation assay (Fig. 3a). As a tail tube protein (TTP) (WP_010328117) from phage SPR was documented to robustly activate the NADase function of DSR2[12], we purified and included the TTP protein in the degradation assay. The results show that full-length DSR2, but not the SIR2 domain-only protein, efficiently cleaved NAD$^+$ in the presence of TTP (Fig. 3a), implying that the CTD region may modulate the NADase activity of the SIR2 domain. Indeed, while tagged TTP failed to pull down the SIR2 domain-only protein, the full-length DSR2 and CTD-only proteins could bind TTP (Fig. 3b).

### Cryo-EM structure of TTP

We next determined the cryo-EM structure of TTP at a resolution of 3.1 Å (Supplementary Fig. 7 and Supplementary Table 1). The quality of the cryo-EM map allowed us to build the atomic model of TTP. The majority of TTP could be unambiguously built, with the exception of the C-terminal portion (aa 240–264), possibly due to intrinsic flexibility. TTP proteins polymerize into a tube

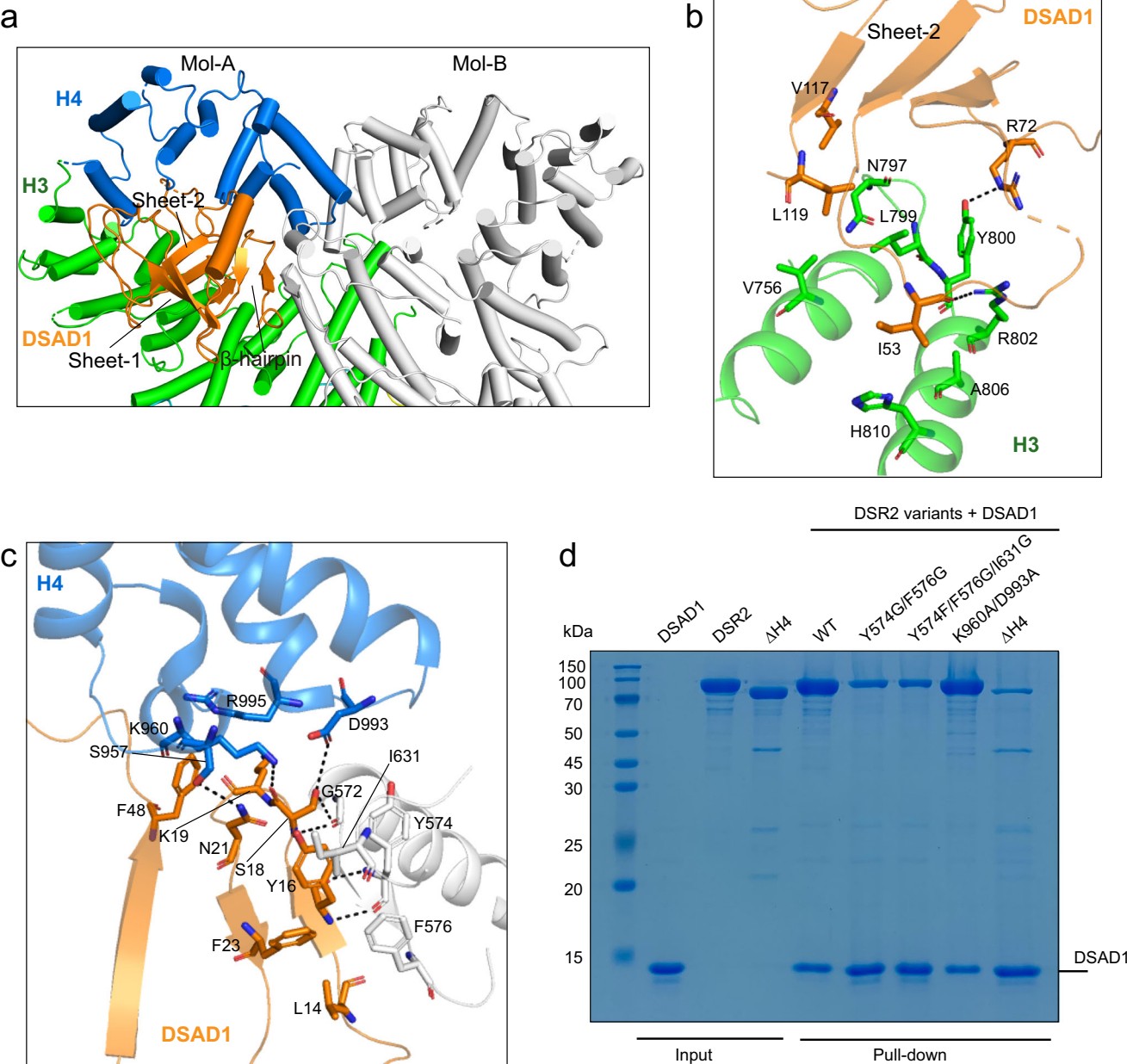

**Fig. 2 | The dimerization of CTD is required for DSAD1 binding. a** Close-up view of DSAD1 binding site on the C-terminal domain (CTD) of DSR2. The DSR2-A molecule is colored in the same scheme as in Fig. 1a, and the DSR2-B molecule is colored in gray for clarity. The DSAD1 protein is colored in orange. **b** Detailed insights into the interaction between DSAD1 and the H3 subdomain of DSR2. Key interacting residues are shown in stick representation. **c** Detailed insights into the interaction between DSAD1 protein and the H3 and H4 subdomains of DSR2 dimer. Key interacting residues are shown in stick representation. **d** In vitro pull-down of wild-type (WT) DSR2 and mutants by His-tagged DSAD1. ΔH4 indicates the deletion of H4 subdomain (aa 1–860). The K960A/D993A mutation had little impact on the DSR2-DSAD1 association. The gel represents three independent replicate experiments. Source data are provided as a Source data file.

with a diameter of ~50 Å (Supplementary Fig. 8a). In each layer of the TTP tube, six TTP molecules are stacked in a head-to-tail arrangement, forming a hexameric ring structure (Fig. 3c). The inner surface of the tube is highly negatively charged (Supplementary Fig. 8b), reminiscent of those observed in *Siphoviridae* phage tails (such as the TTPs of phages YSD1, λ and T5)[25–28], consistent with the proposed genome delivery role of TTP[29]. Each TTP subunit adopts an extended conformation with two β-sandwich domains (termed BS1 and BS2, respectively) (Fig. 3c). The BS1 domains oligomerize into the hexameric ring, which is decorated by the BS2 domains at the periphery. In particular, two β-strands (aa 159–163 and 203–209) of BS1 interact in an antiparallel fashion with the "tail" of the two-layered β-sandwich fold from the neighboring

BS1. Notably, one helix (aa 68–78) is docked into a groove created by these two β-strands and the "head" of the BS1 core, forming a network of hydrophobic intramolecular interactions (Supplementary Fig. 8c), suggesting the importance of this helix (α1) in TTP assembly. The BS2 domain packs against the BS1 domain of the adjacent molecule, further stabilizing the hexameric ring. A β-hairpin (aa 36–56) extending from the BS1 domain interlocks the hexameric rings to form the tube (Fig. 3c). Together, three major intermolecular interfaces, including the two β-strands, BS2 and a β-hairpin-mediated interfaces, facilitate the TTP assembly. The BS1 core of TTP in our study is structurally homologous to TTP from phages YSD1, λ and T5[25–28] (Supplementary Fig. 8d), implying a conserved mechanism for ring formation. Nevertheless, the accessory domains decorating the

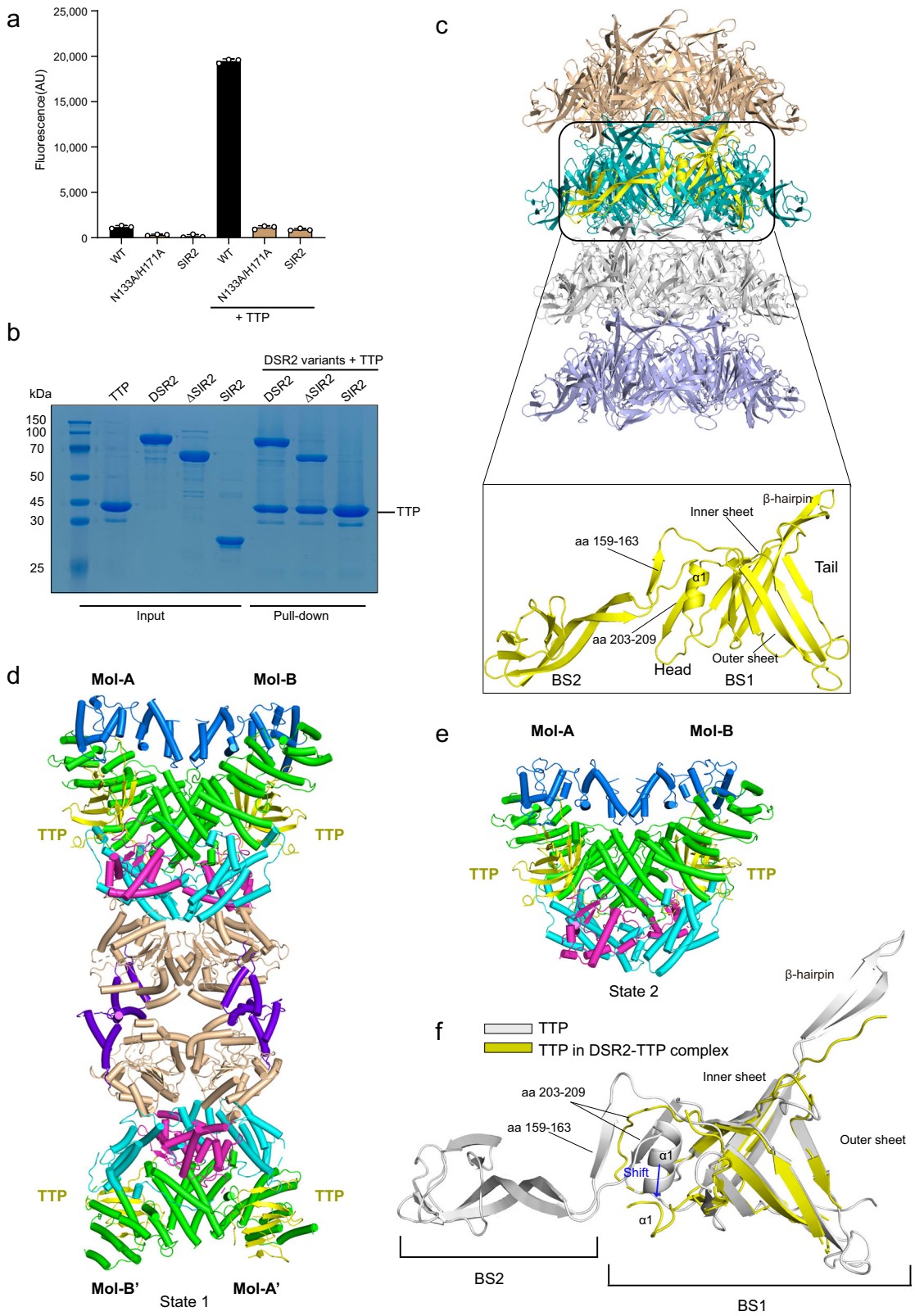

outside of the hexameric ring vary among *Siphoviridae* phage tails. Notably, in phages SPR and YSD1, a β-sandwich domain (BS2 or domain 2) extends from the BS1 core, wrapping around the adjacent TTP subunit. By contrast, immunoglobulin (Ig)-like domains, implicated in host surface molecule recognition, are present in the C-terminal TTP of phage YSD1, λ and T5, but absent in the TTP of phage SPR.

## Cryo-EM structure of DSR2-TTP complex

To elucidate the activation mechanism, we reconstituted the DSR2-TTP binary complex and analyzed its structure by cryo-EM (Fig. 3d, e and Supplementary Fig. 9) (Supplementary Table 1). An elongated DSR2 tetramer binds four TTP molecules (termed state 1) (Fig. 3d), each of which is nestled into a pocket primarily formed by H2-H4 subdomains and an additional H3 subdomain from the neighboring

**Fig. 3 | Cryo-EM structures of TTP alone and in complex with DSR2. a** In vitro NAD+ degradation assays of the full-length DSR2 protein and SIR2 domain-only protein in the presence or absence of TTP. The N133A/H171A catalytic mutant of the DSR2 protein was used as the negative control. 50 μM ε-NAD+ was used as the substrate. Only the WT DSR2 protein can degrade NAD+ in the presence of the TTP protein. The experiment was replicated three times. All experiments were replicated at least three times (mean ± SD, n = 3 independent replicates). **b** In vitro pull-down of wild-type (WT) DSR2 and mutants by His-tagged TTP. The gel represents three independent replicate experiments. ΔSIR2 indicates the deletion of the SIR2 subdomain (aa 304–1005), and SIR2 indicates a SIR2 domain-only protein (aa 1–303). The experiment was replicated three times. **c** Atomic model of the TTP tube

(upper panel), each layer of the TTP ring is colored in wheat, gray, teal, and light blue, respectively. The lower panel shows the structure of TTP subunit in the TTP tube. The single copy of the TTP subunit is shown as cartoon representations and colored in yellow. BS1, β-sandwich domain 1; BS2, β-sandwich domain 2. **d** Cryo-EM structure of DSR2-TTP complex in state 1. The same color scheme as in Fig. 1a is adopted. The four TTP molecules are colored in yellow. **e** Cryo-EM structure of DSR2-TTP complex in state 2. The SIR2 domains are missing in this state. **f** Structural superposition of TTP proteins in the DSR2-TTP complex (yellow) and TTP tube (gray). A shift in the α1 helix of TTP is indicated by a blue arrow. Source data are provided as a Source data file.

molecule (Fig. 3d). Although the DSR2 and the BS1 domain of TTP could be resolved in the EM map (Fig. 3f), the BS2 and the protruding β-hairpin of TTP could not be built, indicating flexibility (Supplementary Fig. 10a). The four DSR2-TTP copies in the heterooctamer are nearly structurally identical. 3D classification also revealed another population of particles showing the missing SIR2 domains (Fig. 3e and Supplementary Fig. 9), which may represent a different state, referred to as state 2 (to be discussed below).

The outer sheet of the BS1 domain pairs with a stretch of residues (aa 903–910) in the H4 subdomain (Fig. 4a, b), while the inner sheet packs over the H3 subdomain. The cleavage of NAD+ was substantially impaired upon the deletion of the H4 subdomain (Fig. 4c). Interestingly, the neighboring H3 subdomain in DSR2 dimer also engages TTP. Specifically, A stretch of disordered region (aa 570–581) refolds into an ordered structure, including a β-strand, which interacts with the inner sheet of BS1 domain upon TTP binding (Fig. 4d), reminiscent of the observation in DSR2-DSAD1 complex (Fig. 2c). Two aromatic residues from the transition region, Tyr574 and Phe576, make hydrophobic and π-π stacking interactions with Ala9, Ala27, Ala30, Phe32 and Tyr175 from TTP (Fig. 4d). Glycine mutation of the two aromatic residues in DSR2 (Y574G/F576G) almost completely abolished NADase activity (Fig. 4c), indicating the critical role of these residues in TTP recognition. In addition, a loop segment (aa 493–504) of H2 transits into an α-helix in the presence of TTP (Fig. 4e). This α-helix of H2 subdomain fits snuggly into a pocket between the two slices of BS1, forming multiple hydrophobic contacts via Leu495, Leu497 and Leu498 (Fig. 4e). Glycine substitution of the leucine residues (L495G/L497G/L498G) abrogated the DSR2-mediated NAD+ cleavage (Fig. 4c). In addition, an extended loop of BS1 (202–213) penetrates a pocket between H2 and H3 subdomains, further reinforcing the interactions around H2 (Fig. 4f). Together, our data demonstrate the molecular basis of TTP recognition by DSR2.

## The CTD acts as a sensor to regulate the SIR2 domain
Strikingly, the DSR2-TTP interface largely overlaps with the contact surface between DSR2 and DSAD1 (Fig. 5a), suggesting that DSAD1 may compete with TTP to bind DSR2. As expected, NAD+ hydrolysis was impeded when simultaneously incubating DSR2 with DSAD1 and TTP (Supplementary Fig. 10b). Notably, DSAD1 showed inhibitory effects regardless of the mixing orders, implying a binding preference of DSR2 for DSAD1 over TTP. It is noteworthy that TTP, in contrast to DSAD1, specifically interacts with the H2 subdomain (Figs. 2a and 4a). This difference may provide a potential explanation for why TTP, but not DSAD1, activates DSR2. The BS1 part of TTP is mainly responsible for contacts with H2 subdomain, which in turn directly makes interaction with SIR2 domain. In the TTP-free state, Tyr148 of SIR2 domain is buttressed by Tyr471, Met531, and Pro532 of H2 subdomain (Fig. 5b). Nonetheless, the inter-domain contacts between H2 and SIR2 appear to be weakened by the binding of TTP (Fig. 5b). Mutation of these residues (M531G/P532G) in the H2 subdomain self-activated the NADase activity of DSR2, even in the presence of DSAD1 (Fig. 5c), further supporting the notion that the H2 subdomain of CTD governs the enzymatic activity of SIR2 domain. Notably, the structure of DSR2-

TTP in state 2 is essentially identical to that in state 1 (Supplementary Fig. 10c), with the exception that the SIR2 domain in state 2 could not be traced, possibly owing to flexibility. Similarly, the SIR2 domain in the APAZ-SIR2/Ago antiviral defense system is also found to be highly dynamic upon recognition of the invading target DNA[16]. We also designed the mutations that are expected to disrupt the interactions between SIR2 and Ago based on the AlphaFold structure[30] (Supplementary Fig. 11a). As anticipated, the mutation could efficiently degrade the NAD+ in the absence of the gRNA-ssDNA duplex (Supplementary Fig. 11b). Therefore, it is conceivable that SIR2-binding regions, such as the H2 subdomain in DSR2 and Argonaute protein in APAZ-SIR2/Ago system, are responsible for transmitting the phage signal to the SIR2 domain. The two functional components of DSR2, CTD, and SIR2, work as the sensor and effector modules in response to phage infection, respectively, which is reminiscent of the anti-phage Cap protein in the CBASS immune system[31,32].

## The lid region of SIR2 domain modulates the NADase activity of DSR2
The overall structure of the DSR2-TTP complex closely resembles that of the DSR2-DSAD1 complex (Fig. 5a), indicating that the binding of either the inhibitor or activator may not elicit substantial conformational changes in the DSR2 tetramer. However, slight structural differences could be observed in the H4 subdomain. Specifically, the H4 subdomain in the TTP-bound molecule displays a closed conformation, analogous to that in the DSAD1-free molecule, but different from the conformation observed in the DSAD1-bound molecule (Fig. 5a). This observation can be attributed to differences in the interaction interfaces, wherein it is DSAD1, rather than TTP, that predominantly engages with the H4 subdomain. In addition to the H4 subdomain, another noticeable difference is the SIR2 domain. Although the overall architecture of SIR2 domain in TTP-bound state is analogous to those in the DSAD1-bound and-free states, the lid region of SIR2 domain undergoes a conformational change (Fig. 5d). Upon binding to TTP, the lid region shifts away from the active-site, possibly exposing the catalytic pocket for substrate binding. Indeed, structural comparison with the structural homolog revealed that the catalytic pocket of SIR2 in DSR2 is likely to accommodate NAD+ after the binding of TTP[33] (Supplementary Fig. 12a). The movement is also likely to impair the inter-dimer interface, leading to the flexibility of the lid region (Fig. 5d). Indeed, the loop regions of the lid are unresolved in the presence of TTP. Mutation of the residues in the lid region responsible for inter-dimer association (Y71A/I90A and replacement of aa 78–96 with a GSAGSA linker) abrogated the NADase function of DSR2 in the presence of TTP (Figs. 1d and 5e), indicating that the lid region may modulate the catalytic activity of SIR2 domain. Together, it is reasonable to conclude that the TTP binding stimulates the rearrangement of the lid region, thereby activating the NADase function of the SIR2 domain. Of note, it has been demonstrated that the closed-to-open transition of the lid region in the SIR2 domain is required for NADase activity, exemplified by the ThsA protein in the Thoeris anti-phage system[11,18]. Structural comparisons between SIR2 domains further confirmed that this activation

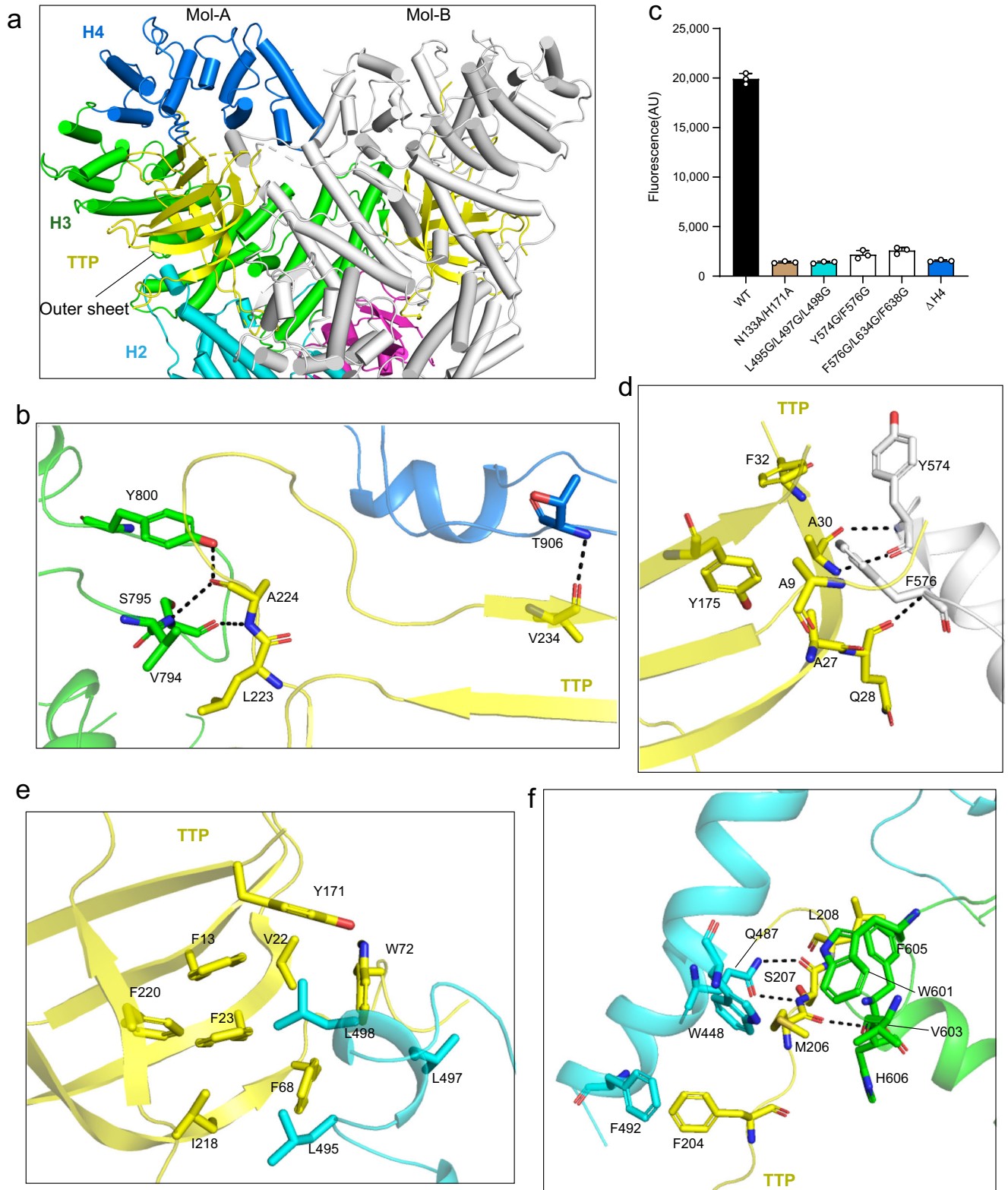

**Fig. 4 | Interactions between DSR2 and TTP proteins. a** Close-up view of the interactions between the BS1 domain of TTP protein and DSR2 proteins. The same color scheme as in Fig. 1a is used for the DSR2-A molecule. The DSR2-B molecule is colored in gray and the bound TTP is colored in yellow. **b** Detailed insights into the interactions between TTP and the H3-4 subdomains of DSR2. Key interacting residues are shown in stick representation. **c** In vitro NAD⁺ cleavage assay using WT or mutants of DSR2 protein. Mutations of the key residues in DSR2-TTP binding interface remarkably reduced NAD⁺ consumption. All experiments were replicated

at least three times (mean ± SD, *n* = 3 independent replicates). **d** Detailed insights into the interactions between TTP and the H3 subdomain of DSR2-B. Key interacting residues in this interface are shown as sticks. **e** Detailed insights into the interactions between TTP and the H2 subdomain of DSR2. Key residues involved in the hydrophobic interaction between TTP and the H2 subdomain of DSR2 are shown in stick representation. **f** Close-up view of the interactions between the TTP and the H2-3 subdomains of DSR2. Key interacting residues are shown in stick representation.

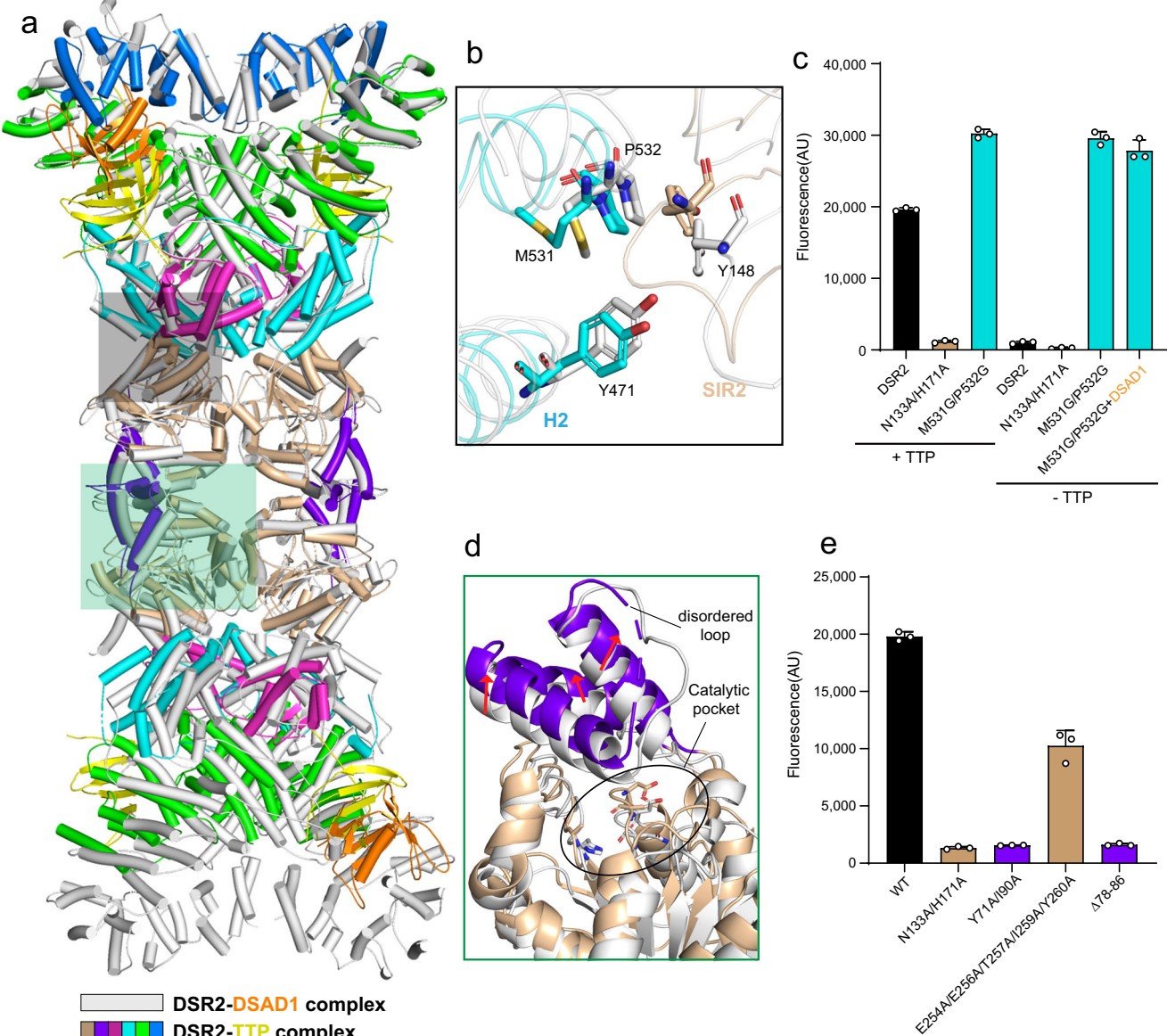

**DSR2-DSAD1 complex**
**DSR2-TTP complex**

**Fig. 5 | The molecular basis of DSR2 activation by TTP protein. a** Structural comparison between the DSR2-TTP complex and DSR2-DSAD1 complex. In the structure of the DSR2-DSAD1 complex, the DSR2 and DSAD1 are colored in gray and orange. The color scheme for the DSR2-TTP complex is consistent with that in Fig. 3d. **b** Close-up view of the inter-domain interaction between the H2 subdomain and SIR2 domain. **c** In vitro NAD+ degradation assay of WT and mutant DSR2 proteins in the presence or absence of TTP. The NADase activity of DSR2 was self-activated through mutations of key residues in the H2-SIR2 interface. All experiments were replicated at least three times (mean ± SD, *n* = 3 independent replicates). **d** Compared to the structure of DSR2 in the DSAD1-binding state, the lid region of the SIR2 domain undergoes conformational changes in the TTP-binding state, leading to outward movement and flexibility in the lid region, thus exposing the catalytic pocket (marked with a black circle). **e** In vitro NAD+ degradation assays investigating the impact of lid region mutations on the NADase activity of DSR2. All experiments were replicated at least three times (mean ± SD, *n* = 3 independent replicates).

mechanism is also applicable to DSR2 (Supplementary Fig. 12b, c). Both Thoeris and APAZ-SIR2/Ago are two-gene operon anti-phage system, whereas it seems that DSR2 alone is sufficient to respond to phage attacks. DSR2 has a two-domain architecture, with the CTD responsible for sensing the infection signal and SIR2 acting to prevent phage propagation (Fig. 6). Our study may also shed light on other single-gene systems, such as DSR1.

## Discussion

Depletion of metabolic substrates, such as NAD+, has been postulated as a mechanism for anti-phage defense in bacterial immunity. The recently identified DSR2 protein exerts its anti-phage activity through its NADase function. Here, we investigated the molecular basis underlying the inhibition and activation of DSR2 by phage

proteins (Fig. 6). Two DSR2 dimers are assembled in a head-to-head fashion by the tetramerization of SIR2 domains (Fig. 1b), which is analogous to the observation in the SIR2-containing ThsA protein of the Thoeris system[11,18]. In contrast, the APAZ-SIR2/Ago complex, another anti-phage system with NADase activity, functions as a monomer[7,16]. While the oligomerization states of SIR2-containing effectors vary among anti-phage systems, the activation mechanism, involving an ordered-to-disordered transition of the lid region, is conserved. Intriguingly, another NADase domain, TIR, is also implicated in the anti-phage defense. However, oligomerization of TIR domain is required for the formation of the composite active-site[16,34–42], which contrasts with the finding in SIR2-containing proteins, such as DSR2 and ThsA, where destabilization or dissociation of SIR2 assembly is indispensable for NADase function.

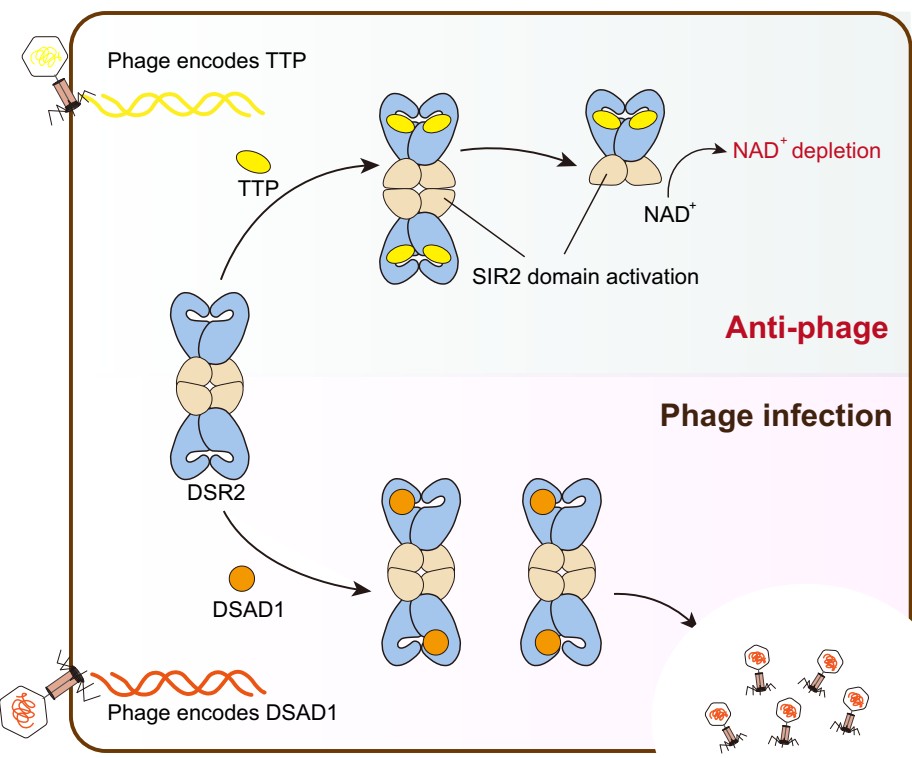

**Fig. 6 | Schematic diagram illustrating the NADase activation and inhibition mechanisms of DSR2.** The TTP binds to DSR2 and allosterically activates the SIR2 domain, while the DSAD1 protein encoded by some phages mimics TTP, occupying the same pocket in DSR2 and thereby inhibiting NADase activity.

It has been demonstrated that the TTP from phage SPR, but not phage SPbeta, efficiently binds and activates DSR2[12]. Importantly, it is noteworthy that several residues crucial for the interaction with DSR2 are not conserved in TTP. For instance, in the TTP of phage SPR, Phe13 and Phe23 are involved in hydrophobic contacts with DSR2, while the equivalent residues are Ala15 and Ala25 in the TTP of phage SPbeta. This could explain why DSR2 is specifically activated by TTP of phage SPR. The assembly of TTP is essential for the delivery of the phage genome. Although the BS1 core of TTP is not altered remarkably in the presence of DSR2, the three primary interfaces that mediate TTP assembly, however, are disrupted upon binding to DSR2. Particularly, the extended β-hairpin in TTP, which is involved in stacking the hexameric rings (Fig. 3c), plugs into the H4 subdomain of DSR2 (Fig. 4a), thereby potentially impeding the formation of tubal structure (Supplementary Fig. 13). Furthermore, the α1 helix of TTP, which is implicated in maintaining the two β-strands-mediated hexameric ring assembly (Fig. 3c and Supplementary Fig. 7c), shifts away from the two β-strands-involved groove (Fig. 3f), possibly destabilizing TTP assembly. Indeed, the two β-strand regions of TTP are flexible in the presence of DSR2 (Fig. 3f). Therefore, it seems that the DSR2 binding would impair tube formation, mitigating phage infection.

Remarkably, the inhibitor DSAD1 and the activator TTP occupy the same pocket formed by the H3 and H4 subdomains of DSR2 (Fig. 5a). In addition, a similar set of residues is employed by DSR2 to interact with both DSAD1 and TTP. For instance, two aromatic residues (Tyr574 and Phe576) from the H3 subdomain are involved in the binding of both DSAD1 and TTP (Figs. 2c and 4d). Therefore, it is plausible that DSAD1, at least to some extent, binds to DSR2 by mimicking the TTP. There are substantial steric clashes at both the DSR2-DSAD1 and the DSR2 dimerization interfaces when aligning the DSAD1-free DSR2 molecule (Mol-B) with the DSAD1-bound DSR2

(Mol-A), potentially explaining why DSAD1 occupies only two binding sites in the context of NADase inhibition (Supplementary Fig. 14a). Similarly, remarkable spatial clashes are observed at both the DSR2-DSAD1 and the DSR2 dimerization interfaces when structurally superimposing the TTP-bound DSR2 (Mol-A) onto the DSAD1-free DSR2 molecule (Mol-B) (Supplementary Fig. 14b). Therefore, it is reasonable to conclude that the ligand-free pockets in DSR2 of the DSR2-DSAD1 complex are not potentially available for additional DSAD1 or TTP binding. However, further study is required to clarify the binding of ligands to DSR2, particularly at higher concentrations.

## Methods
### Protein expression and purification
The gene sequences encoding DSR2 (WP_029317421), DSAD1 (WP_004399562), and TTP (WP_010328117) were synthesized (Genewiz) and cloned into expression vectors. The full-length and truncated DSR2 were cloned into the 2CT-10 vector (Addgene plasmid #55209) with a His-MBP affinity tag and a TEV recognition site at the N-terminal of the protein. TTP, DSAD1, and DSR2-TTP were cloned into the pET-derived vector with an N-terminal His tag.

For expression and purification, the plasmids were transformed into *E. coli* Rosetta (DE3) competent cells. Protein expression was induced by adding 0.2 mM isopropyl-β-D-thiogalactoside (IPTG) to the culture and shaking for 16 h at 16 °C. Cells were pelleted after induction and resuspended in the binding buffer (25 mM Tris-HCl pH 7.5, 500 mM NaCl, 5 mM imidazole, and 3 mM β-mercaptoethanol), and lysed by sonication. The lysates were then incubated with Ni-NTA resin (QIAGEN) and the beads were extensively washed with binding buffer supplemented with 20 mM imidazole. The proteins were eluted with binding buffer containing 300 mM imidazole. The eluate was incubated with TEV protease at 4 °C overnight to remove the His-MBP tag.

Anion-exchange chromatography (HiTrap Q HP, Cytiva) followed by gel filtration chromatography (Superdex 200 Increase, Cytiva) was used for further purification. Peak fractions were concentrated and stored in gel-filtration buffer (25 mM Tris-HCl pH 7.5, 150 mM NaCl, 2 mM DTT) before use. The protein complexes were obtained by co-expression and co-purification methods. Protein purity was analyzed by SDS-PAGE.

## Analytical ultracentrifugation

The sedimentation velocity of tested proteins was measured using a Beckman Optima XL-I analytical ultracentrifugation. Samples were diluted to 1 mg/mL in gel-filtration buffer before use. Sedimentation coefficient distribution was calculated with SEDFIT and SEDPHAT programs[43,44].

## ε-NAD⁺ degradation assays

The WT or mutant DSR2 proteins were incubated with TTP at 37 °C for 30 min. The substrate $\varepsilon$-NAD$^+$ (Sigma-Aldrich) was then added to initiate the reactions in a buffer containing 25 mM Tris-HCl, pH 7.5, 150 mM NaCl, and 5 mM MgCl$_2$. The final concentrations were 1 μM, 5 μM, and 50 μM for DSR2 variants, TTP and $\varepsilon$-NAD$^+$, respectively. After incubation at 37 °C for 2 h, the reactions were measured on a BioTek Synergy H1 plate reader ($\lambda_{ex}$, 310 nm; $\lambda_{em}$, 410 nm). To test the inhibitory effect of DSAD1 on the NADase activity of DSR2, 5 μM DSAD1 and 5 μM TTP were incubated with DSR2 in different orders. All assays were performed in triplicate. The means and standard deviations were calculated using GraphPad Prism v.8.3.

## His pull-down assay

The N-terminal His-tagged TTP or DSAD1 proteins were incubated with 50 μL Ni-NTA agarose beads for 30 min at 4 °C in binding buffer (25 mM Tris-HCl pH 7.5 and 300 mM NaCl). Excessive untagged DSR2 variants were added to the bait protein and incubated with the resin for 1 h at 4 °C. The beads were then washed four times with 1 mL binding buffer supplemented with 10 mM imidazole to remove the unbound proteins. The bound fractions were then eluted with a buffer containing 500 mM imidazole. The samples were resolved on 12% SDS-PAGE and the gels were stained by using Coomassie Brilliant Blue. Three independent replicates were performed.

## Native polyacrylamide gel electrophoresis

Purified DSR2 protein was diluted to 1 mg/mL in gel-filtration buffer and loaded onto a 5% native gel. The electrophoresis was performed at 120 V for 40 min at 4 °C after a pre-run at 4 °C for 30 min. Protein bands were visualized with Coomassie Brilliant Blue staining. The experiment was repeated three times.

## Cryo-EM data collection

Aliquots of 4 μL protein complexes (0.5 mg/ml) were applied to glow-discharged Quantifoil holey carbon girds (Au, R1.2/1.3, 300 mesh). The grids were blotted with force 2 for 8 s and plunged into liquid ethane using Vitrobot. Cryo-EM data were collected with a Titan Krios microscope (FEI) operated at 300 kV and images were collected using EPU[45] at a nominal magnification of ×105,000 (resulting in a calibrated physical pixel size of 0.85 Å/pixel) with a defocus range from −1.2 to −2.2 μm. The images were recorded on a K3 summit electron direct detector in super-resolution mode at the end of a GIF-Quantum energy filter operated with a slit width of 20 eV. A dose rate of 15 electrons per pixel per second and an exposure time of 2.5 s were used, generating 40 movie frames with a total dose of ~54 electrons per Å$^2$. A total of 3149, 3547, and 1648 movie stacks were collected for DSR2-TTP complex, TTP tube, and DSR2-DSAD1 complex, respectively (Supplementary Table 1).

## Cryo-EM data processing

Cryo-EM data processing for the DSR2-DSAD1 complex was initially performed with cryoSPARC[46] and RELION-3[47]. 1648 movie frames were aligned using MotionCor2[48] with a binning factor of 2. Contrast transfer function (CTF) parameters were estimated using Gctf[49]. Around 15,000 particles were auto-picked without template to generate 2D averages for subsequent template-based auto-picking using RELION-3[47]. 1,626,566 particles were auto-picked and extracted from the dose-weighted micrographs. 2D classification was performed to exclude false and bad particles that fall into 2D averages with poor features. 666,848 particles were selected for further processing. 3D classification was performed to distinguish different conformational states. 307,891 particles were used for final 3D refinement, converging at 3.44 Å resolution. In addition, Blob picker and Topaz in cryoSPARC[46] were also utilized for particle picking. A total of 228,964 particles were automatically selected and extracted from the micrographs. These particles were subsequently subjected to 2D classification. Good classes were selected after two rounds of ab initio reconstruction and heterogeneous refinement. Two classes, with a total of 96,511 particles, were chosen for the final 3D refinement, resulting in a final reconstruction with an overall resolution of 3.59 Å. Another class, with a total of 90,023 particles, was also selected for the final 3D refinement, resulting in a final reconstruction with an overall resolution of 3.59 Å.

Cryo-EM data sets of TTP tube were processed using cryoSPARC[46]. The patch CTF estimation of 3547 micrographs were performed with cryoSPARC[46]. 526 segments of TTP filament were manually picked and sent to 2D classification, two good classes were selected as templates for the automated filament tracer in cryoSPARC[46]. A 120 Å filament diameter and 0.3 diameters between segments were used for template-based filament tracer. 199,236 segments were extracted with a box size of 320 pixels. 2D classification was performed to exclude false and bad particles that fall into 2D averages with poor features. 116,902 particles were selected for ab initio helical refinement, employing C6 symmetry, a rise of 42.4 Å, and a twist of 18.5°. The refinement process resulted in a reconstructed map with a resolution of 3.11 Å.

Cryo-EM data of the DSR2-TTP protein complex were processed using RELION-3[47]. Movie frames were aligned using MotionCor2[48] with a binning factor of 2. Contrast transfer function (CTF) parameters were estimated using Gctf[49]. Around 15,000 particles were auto-picked without a template to generate 2D averages for subsequent template-based auto-picking. 2,957,176 particles were auto-picked and extracted from the dose-weighted micrographs. 2D classification was performed to exclude false and bad particles that fall into 2D averages with poor features. 1,296,749 particles were selected for further processing. Particles from different views were used to generate the initial model in cryoSPARC[46]. 3D classification was performed to distinguish different conformational states. 725,557 particles were used for final 3D refinement with C2 symmetry imposed, converging at 2.95 Å resolution. Further 3D classification on the dataset revealed a minor population containing 20,085 particles, and they were used for final 3D refinement, resulting in a resolution of 4.48 Å resolution. Details of the cryo-EM image processing is summarized in Supplementary Table 1.

## Model building and refinement

Atomic models of DSR2, TTP, and DSAD1 were predicted by the AlphaFold pipeline in ColabFold[30,50] and manually docked into the cryo-EM maps using ChimeraX[51], followed by the manual adjustment and refinement using Coot[52]. Further iterative refinement was carried out using *phenix.real_space_refine*[53] and Coot[52]. Model validation was performed using MolProbity[54] in PHENIX[55]. The detailed data processing and structure refinement statistics are summarized in Supplementary Table 1.

## Reporting summary

Further information on research design is available in the Nature Portfolio Reporting Summary linked to this article.

## Data availability

The atomic coordinates and EM maps have been deposited in the Protein Data Bank under accession codes 8WKN (DSR2-DSAD1), 8W56 (DSR2-DSAD1 state 1), 8K9A (DSR2-DSAD1 state 2), 8XKN (TTP), 8WFN (DSR2-TTP state 1), 8K98 (DSR2-TTP state 2), and in the Electron Microscopy Data Bank under corresponding accession codes EMD-37603, EMD-37272, EMD-36982, EMD-38421, EMD-37497, and EMD-36980. Source data are provided with this paper.

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

## Acknowledgements

We thank the staff members from Cryo-Electron Microscopy Facility of Hubei University for help on data collection and computation. We thank Dr. Qin Cao for helpful suggestions. This work was supported by the National Natural Science Foundation of China (32322040 to H.Z., 32300036 to H.Y. and 32201004 to Z.L.), National Key R&D Program of China (2022YFC3400400 to Y.W. and 2022YFA0911800 to Z.L.), the Hubei Provincial Natural Science Foundation (ZRMS2022000096 to Z.L.) and Scientific Research Program of Tianjin Municipal Education Commission (2022KJ192 to H.Y.).

## Author contributions

H.Z. supervised the project; H.Z., Z.L., Y.W. and H.Y. designed the project; H.Y., X.L., X.W., J.G., Q.H., J.Y. and X. Liu performed molecular cloning, protein purification, and biochemical assays; C.Z., X.L., Z.L. and G.Y. performed the cryo-EM works; H.Z. wrote the manuscript with contributions from the other authors.

## Competing interests

The authors declare no competing interests.
