## [Peer Review File · Nature Communications]

Reviewers' Comments:

Reviewer #1:

Remarks to the Author:

The authors have addressed all the issues I previously raised.

Here I only have minor points.

1. Suggest citing references in two places:

Lines 50-60 "NAD+ is crucial for cellular metabolism."

Lines 86-87 "....., supporting the emerging paradigm of cooperative self-assembly formation in immune signaling pathways."

2. Line 207, Siphoviridae should be italic.

3. In Figure 6, the different cartoon presentation of DSAD1 in the two states is confusing.

Reviewer #2:

Remarks to the Author:

The authors have revised the manuscript to address my and the other reviewer's concerns in a comprehensive manner. Crucially, they have added new figures with comparisons of their structures with existing structures and new insights to the results and discussion sections. The quality of their models has been improved in regions of poor density, and model-map fits have been provided in the supplementary material. The authors have also reprocessed some of the cryo-EM data using the correct symmetries, which I was pleased to see resulted in a more consistent resolution for the TTP and improved the resolution for the DSR2-DSAD1 reconstructions. The authors also explained why symmetry averaging was not beneficial for some of the reconstructions, and illustrated how the observed conformations of DSR2 can permit or occlude TTP/DSAD1 binding.

I have some minor comments and suggestions for the revised material listed below:

"To understand the assembly mechanism, we sought to solve the structure of DSR2 alone by cryo-EM. However, we could not obtain a high-resolution 3D reconstruction, possibly due to intrinsic flexibility." – I saw that you added the 2D class from this data in S4a, but you have neglected to mention it here.

"RE: We have included the particle picking in the revised figures." – I acknowledge you have done so for Figures S7/S9, but not in Figure S2. The information/labelling provided for the workflows in S2/S7/S9 are inconsistent e.g. number of micrographs, number of particles before 2D classification, exposure curation etc. appear in some but not others.

There are some minor grammatical errors in the new sentences added to the manuscript e.g. "such as TTP in phage YSD1, λ and T5" -> "such as the TPPs of phages YSD1, λ and T5"

Figure S4a legend - missing description of the right-hand image.

Figure S7b – label under 2D classes reads 'Particles from helical segment picker', but these are 2D classes, not particles

Figure S7 has a duplication of the figure legend (I assume the second instance is the correct one).

Figure S8b - missing units for the electrostatics charge

Figure S8d – explain in the legend that there are two alignments shown for T5, one for each BS1 domain

Figure S9 – please swap panels 'h' and 'i', so that they match the order of 'd' and 'f' as indicated in the legend.

Table S1 - Please indicate that you used helical symmetry as well as C6 symmetry for the TTP reconstruction

Reviewer #3:

Remarks to the Author:

In general the authors have addressed my comments satisfactorily. The figures are now much improved and the manuscript is acceptable for publication.

However, I still feel that it is very difficult to see how the highlighted side chains in the various figures actually relate to the main chain backbone; they almost appear to be floating in space. Could I suggest that the authors try to use ChimeraX to create the figures combining cartoon representation with stick representation of selected side chains.

Reviewer #1 (Remarks to the Author):

The authors have addressed all the issues I previously raised.

RE: We are grateful for the reviewer's insightful comments for improving our manuscript.

Here I only have minor points.

1. Suggest citing references in two places:

Lines 50-60 "NAD+ is crucial for cellular metabolism."

Lines 86-87 "..., supporting the emerging paradigm of cooperative self-assembly formation in immune signaling pathways."

RE: Thanks for the suggestions, we have cited the references as suggested.

2. Line 207, Siphoviridae should be italic.

RE: We amended the text as suggested.

3. In Figure 6, the different cartoon presentation of DSAD1 in the two states is confusing.

RE: We apologize for the confusion, and we have remade the figure.

Reviewer #2 (Remarks to the Author):

In general the authors have addressed my comments satisfactorily. The figures are now much improved and the manuscript is acceptable for publication.

RE: We appreciate the reviewer's constructive feedback, which has helped us strengthen the manuscript.

However, I still feel that it is very difficult to see how the highlighted side chains in the various figures actually relate to the main chain backbone; they almost appear to be floating in space. Could I suggest that the authors try to use ChimeraX to create the figures combining cartoon representation with stick representation of selected side chains.

RE: We apologize for the confusion. The floating chains arose from the cylindrical helix representation in PyMOL. We have addressed this issue by changing the presentation of helices, and floating chains are no longer present in the new figures.

Reviewer #3 (Remarks to the Author):

The authors have revised the manuscript to address my and the other reviewer's concerns in a comprehensive manner. Crucially, they have added new figures with comparisons of their structures with existing structures and new insights to the results and discussion sections. The quality of their models has been improved in regions of poor density, and model-map fits have been provided in the supplementary material. The authors have also reprocessed some of the cryo-EM data using the correct symmetries, which I was pleased to see resulted in a more consistent resolution for the TTP and improved the resolution for the DSR2-DSAD1 reconstructions. The authors also explained why symmetry averaging was not beneficial for some of the reconstructions, and illustrated how the observed conformations of DSR2 can permit or occlude TTP/DSAD1 binding.

RE: We thank the reviewer for the thoughtful suggestions that have improved our manuscript.

I have some minor comments and suggestions for the revised material listed below:

"To understand the assembly mechanism, we sought to solve the structure of DSR2 alone by cryo-EM. However, we could not obtain a high-resolution 3D reconstruction, possibly due to intrinsic

flexibility.” – I saw that you added the 2D class from this data in S4a, but you have neglected to mention it here.

RE: Thanks for the suggestions, we have mentioned it as suggested.

“RE: We have included the particle picking in the revised figures.” – I acknowledge you have done so for Figures S7/S9, but not in Figure S2. The information/labelling provided for the workflows in S2/S7/S9 are inconsistent e.g. number of micrographs, number of particles before 2D classification, exposure curation etc. appear in some but not others.

RE: Our apologies. We have changed the information in these figures.

There are some minor grammatical errors in the new sentences added to the manuscript e.g. “such as TTP in phage YSD1, λ and T5” -> “such as the TPPs of phages YSD1, λ and T5”

RE: We amended the text as suggested.

Figure S4a legend - missing description of the right-hand image.

RE: We have added the description.

Figure S7b – label under 2D classes reads ‘Particles from helical segment picker’, but these are 2D classes, not particles.

RE: We have corrected the label as suggested.

Figure S7 has a duplication of the figure legend (I assume the second instance is the correct one).

RE: Thanks for pointing this out. We have deleted the incorrect legend.

Figure S8b - missing units for the electrostatics charge.

RE: We have included the units.

Figure S8d – explain in the legend that there are two alignments shown for T5, one for each BS1 domain.

RE: We have included the description as suggested.

Figure S9 – please swap panels ‘h’ and ‘i’, so that they match the order of ‘d’ and ‘f’ as indicated in the legend.

RE: We have swapped these two panels as suggested.

Table S1 - Please indicate that you used helical symmetry as well as C6 symmetry for the TTP reconstruction

RE: We have included the helical symmetry in Table S1.